# Exploring barriers to integrated care for children under 5 living in temporary accommodation: a qualitative study of professionals' experiences during the COVID-19 pandemic in England

Poppy Angelica Spaceman Pierce [1], Nadzeya Svirydzenka [2], Haleema Adil,[1] Shereen Allaham [3], Matthew Ankers [4], Yvonne Karen Parry,[5] Michelle Heys [3,6], Marcella Ucci [7], Monica Lakhanpaul [3,8]

PASP and NS are joint first authors.

For numbered affiliations see end of article.

**Correspondence to**
Professor Monica Lakhanpaul;
m.lakhanpaul@ucl.ac.uk

## ABSTRACT

**Objectives** This research aims to explore the factors that hinder professionals in delivering integrated care for children under 5 in temporary accommodation (TA) and understand their experiences of collaboration during the pandemic to inform recommendations.

**Design** Semistructured qualitative interviews.

**Setting** England, UK.

**Participants** 45 professionals working across health, housing, education and non-profit sectors in England. Purposive and snowball sampling was employed to recruit a representation of key professionals across England. Those not eligible to take part in the study included people who did not work with families and/or children in TA settings.

**Outcome measures** To explore cross-sector learnings that are applicable to improving integrated care and to tailor recommendations to the needs of families and children under 5 experiencing homelessness in the UK today.

**Results** This study highlights the complex, multilevel barriers that professionals face when delivering integrated care to children under 5 in TA. Findings are organised using a framework that distinguishes between practice-level, organisational and systemic challenges. From siloed working practices and limited training to staffing shortages and restrictive data-sharing policies, these challenges collectively hinder service continuity and collaboration.

**Conclusions** Although this project was conducted during the COVID-19 pandemic, the challenges identified reflect deeper, long-standing issues in service delivery. As services continue to recover and prepare for future crises, these insights remain highly relevant and can inform more resilient, integrated recovery plans to support children in TA beyond the pandemic context. Addressing these barriers, through improved collaboration, training and data-sharing, is key to strengthening care for this vulnerable population.

## INTRODUCTION

Family homelessness is an escalating issue within high-income countries, including the USA and across Europe.[1] In the UK, this trend is particularly alarming, with over

---

**STRENGTHS AND LIMITATIONS OF THIS STUDY**

⇒ The study included diverse professionals from health, housing and social care across England, enhancing relevance and transferability.

⇒ Semistructured interviews were conducted with a detailed topic guide, allowing for consistency and rich, nuanced data collection.

⇒ Limited staff availability due to the COVID-19 pandemic led to opportunistic rather than representative sampling.

⇒ The study could not account for local variation in services or enable comparative regional analysis.

---

164 000 children living in temporary accommodation (TA) with their families in 2024; an increase of 15% since 2023.[2] The rising numbers reflect the deepening housing crisis, exacerbated by economic pressures, a lack of liveable housing stock and inadequate access to affordable housing.[3] The 2025 Shelter Scotland report provides stark evidence of how living in TA harms children's health, development and education, reinforcing the urgent need for cross-sector action to prevent long-term consequences.[4] The COVID-19 pandemic saw a sharp increase in the number of families living in TA across England, impacting an estimated 253 000 individuals.[5][6] Children under 5 were particularly susceptible to poor health outcomes, due to limited healthcare access; unstable and substandard living conditions; and the disruption to early development.[7–9] The COVID-19 pandemic inadvertently served as a stress test for public services, showcasing effective practices but also exposing pressing issues within these systems.[10] Addressing these enduring challenges is urgent, as this

is a critical period of rapid growth for children under 5, where timely intervention can prevent long-term developmental challenges and offer a unique opportunity to positively shape a child's future.[7 11 12]

The COVID-19 pandemic significantly disrupted care systems worldwide but also compelled health and social care partners to collaborate more closely and rapidly than they had previously.[13 14] Professionals reported improved cross-organisational working as priorities aligned to address the crisis.[15] Integrated community health responses and initiatives such as 'Everyone In' helped maintain care access and positioned homelessness as a public health priority.[16 17] However, children were largely overlooked in COVID-19 policy and planning, with low-income families with children under 5 being disproportionately disadvantaged.[18 19] Longstanding research has consistently identified structural barriers to effective health, housing and social care provision for homeless families, well before the COVID-19 pandemic. Studies spanning the past two decades highlight persistent challenges including fragmented service delivery, limited cross-sector coordination, workforce instability and bureaucracies across systems supporting both adults and children experiencing homelessness.[20–23] The pandemic highlighted and intensified numerous of these pre-existing vulnerabilities in the systems that support children under 5 in TA, such as staff shortages, limited training and communication breakdown between teams.[12] Following the pandemic, many of these issues remain unresolved, with additional challenges like the cost of living, austerity measures and the housing crisis continuing to hinder recovery.[24–27] The result is overlapping crises which are also likely to be exacerbating each other. As we anticipate future crises, it is imperative to address healthcare provision for children under 5 in TA and how to sustain such provision in a system under stress. The majority of children under 5 in TA face complex and multifaceted challenges, where solutions do not necessarily lie within the remit of a singular organisation.[28 29] Effective cross-sectoral action is needed to meet current demands, and the work to recover, prepare and innovate must be framed through a health equity lens.[29–32]

While previous research has extensively documented systemic barriers to integrated care for homeless populations, less is known about how these long-standing challenges were experienced, adapted to or exacerbated during the COVID-19 pandemic. Although research on children under 5 in TA during the pandemic is scarce, one study conducted in Newham, an exceptionally deprived and diverse inner-city London borough, offers valuable local insight into professionals' perspectives on the barriers families in TA faced in accessing healthcare during this time.[12] While the previous study focused on professionals' perspectives on families' challenges within a single borough, this study shifts the lens to professionals' reflections on their own roles and the barriers to cross-sector collaboration at a national level. In doing so, it builds on and extends existing knowledge, addressing a critical gap in understanding how to improve cross-sector working to better support children under 5 in TA. This research aims to qualitatively explore the factors that hinder professionals in delivering integrated care for children under 5 in TA and understand their experiences of collaboration during the pandemic to inform recommendations. The recommendations follow two key objectives: (1) to explore cross-sector learnings that are applicable to improving integrated care and (2) to tailor recommendations to the needs of families and children under 5 experiencing homelessness in the UK today.

## METHODS

This study sits within the CHAMPIONS programme of research which focuses on the impact of living in poverty on children under 5 living in TA.[33] During the COVID-19 pandemic, a mixed method multiphased study was conducted with the specific aim of understanding the experiences of children in TA during that time. This paper represents only the work package dedicated to the qualitative interviews with professionals engaged with and supporting the cross system around the child and their families.

### Participants

Purposive and snowball sampling was employed[34 35] to recruit a representation of key professionals across England. Those not eligible to take part in the study included people who did not work with families and/or children in TA settings. No participants were excluded based on protected characteristics or any demographic factors. Participants were recruited through a variety of channels, including:

► Housing, welfare, refugee and domestic violence support organisations, as well as housing providers such as local authorities and housing associations.
► Direct referrals from professionals, including health visitors and housing officers.
► Social media platforms and WhatsApp groups managed by the project team and partner support organisations.
► Leaflets and posters were distributed through food banks and family support charities.
► Snowballing through word of mouth.

Recruitment was undertaken between September 2021 and March 2022. Participants were identified professionals within key areas who were invited to take part in the study via an email, leaflet or word of mouth. All potential participants were asked to email the researchers to indicate their interest in taking part. Interested participants received an information sheet and were encouraged to respond with any questions, or to arrange an interview. Informed consent was obtained prior to participation. Research fellow on the project continued with recruitment and data collection until thematic saturation was reached, which occurred after 45 interviews.[36] Despite the challenges of conducting research during the COVID-19

pandemic, the study achieved a strong sample size for qualitative research as data saturation was reached.[37] Effective recruitment was facilitated by strong participant engagement strategies, established trust and the research team's experience and credibility in the field.

## Materials

The interview schedule was developed by looking into key domains of supporting families with children under 5 in TA. To gather detailed insights into the perceived barriers to delivering integrated care during the pandemic, the questions addressed topics such as training related to supporting children under 5 in TA; workload and capacity; workplace dynamics and collaboration between sectors, particularly between health, non-profit and housing services; impact of COVID-19 pandemic on families living in TA; the impact of policies and funding on service delivery; personal approaches to providing care; and recommendations for improving integrated care for children under 5 in TA. The topic guide is presented in online supplemental file 1. A demographic questionnaire was also given to participants to complete prior to the interviews. A brief agenda was sent before the interview outlining the topics to be covered and reassuring participants that they did not have to answer any questions they were uncomfortable with.

KF arranged individual semistructured interviews with participants, conducted remotely via Zoom[38] at a place of the participants' choice, usually their workplace or home office. KF is a research fellow, with a background of a PhD in sociology, MA in social research, MPhil, BA (Hons), with significant experience in conducting qualitative market research interviews. This was disclosed to participants. A relationship with participants and KF was not established prior to the interviews taking place. Participants were introduced to the researcher as a person working on a project with larger goals relating to identifying children's needs post-COVID-19, codeveloping requirements for COVID-19 recovery and reporting them to policymakers and practitioners. All interviews were carried out in English, recorded using Zoom's digital platform. Interviews were scheduled to take no more than an hour but were shorter or longer according to the participants' time available and their convenience. Only the researcher and participant were present at the interview. No repeat interviews were carried out and no field notes were made. Transcription was carried out by the research fellow or professional transcriber and quality cross-checked with the audio file.

The Consolidated criteria for Reporting Qualitative research reporting checklist is provided in online supplemental file 2.

## Analysis

All interview transcripts were anonymised and given a four-digit number code. Researchers PASP and HA carried out data analysis using an inductive approach to generate codes through data familiarisation. They were supported throughout by senior team members, including a paediatrician with in-depth knowledge of health and social care systems and academic researchers with expertise in qualitative methods and the relevant subject matter, which strengthened the rigour and interpretation of the findings. Two interviews were co-coded independently and codes were then compared with ensure replicability and reliability. In cases where conflict in codes arose, consensus discussions took place and codes were revised. In the final stage, findings were organised using a multi-level framework that distinguishes between practice-level, organisational and systemic challenges. Codes were subsequently categorised into sub-themes that emerged in analysis. This followed Braun and Clarke's six-step framework for qualitative analysis.[39] NVivo software (V.12) was used to assist in the organisation and analysis of the data.[40]

## Patient and public involvement

None.

## RESULTS

In total, 45 interviews were conducted. The average time for an interview was 1 hour. Table 1 shows participant characteristics. Professionals included 9 health visitors, 16 health professionals, 10 professionals from non-profit organisations, 8 housing sector professionals and 2 professionals working in education. The participants' ages ranged from 26 to 68 years, with a mean age of 44. The majority were female (39/45), and they had experience working with families living in TA for periods ranging from 1 to over 40 years, with an average of 8 years.

## Roles of participants interviewed

The professionals interviewed in this study represented a diverse range of roles across healthcare, housing, education and the charity sector, all working to support children under 5 living in TA in England. Healthcare professionals included general practitioners, health visitors, midwives, nurses, dentists, a paediatrician and a speech and language therapist, many specialising in homelessness, safeguarding, refugee support and perinatal care. Housing professionals, including caseworkers, support workers, solicitors and housing directors, highlighted that they provided advice on housing rights, challenged poor living conditions and liaised with local authorities. Educators, such as an early years teacher and a nursery director, supported children's development in challenging circumstances. Non-profit professionals, including charity directors, coordinators, peer support leads, play workers, policy advisors and volunteers, highlighted their focus on direct service delivery, emotional and educational support, advocacy and strategic planning to address the complex needs of families in TA.

## Barriers

The results of this study identified a range of barriers affecting professionals' ability to deliver integrated care

**Table 1** Participant characteristics including role, age, gender, ethnicity and length of time in role

| Professional group | Number of professionals n (%) |
| --- | --- |
| Health visitors | 20 (9) |
| Health professionals | 35.6 (16) |
| Non-profit sector | 22.2 (10) |
| Housing sector | 17.8 (8) |
| Education | 4.4 (2) |
| **Age group** | **n (%)** |
| Not given | 4.4 (2) |
| 25–29 | 6.7 (3) |
| 30–34 | 15.6 (7) |
| 35–39 | 13.3 (6) |
| 40–44 | 15.6 (7) |
| 45–49 | 22.2 (10) |
| 50–54 | 2.2 (1) |
| 55–59 | 6.7 (3) |
| 60–64 | 11.1 (5) |
| 65–69 | 2.2 (1) |
| **Gender** | **n (%)** |
| Not given | 2.2 (1) |
| Male | 11.1 (5) |
| Female | 83.7 (39) |
| **Ethnic group** | **n (%)** |
| Not given | 6.7 (3) |
| White (British and/or any other White British background) | 77.8 (35) |
| Black (African or Caribbean) | 6.7 (3) |
| Asian or Asian British: Indian | 6.7 (3) |
| Mixed Ethnic background | 2.2 (1) |
| **How long professionals have worked with families living in temporary accommodation (time spans)** | **n (%)** |
| <1 year | 0 |
| 1–5 years | 26.7 (12) |
| 5–10 years | 33.3 (15) |
| >10 years | 40 (18) |
| Total | 45 (100%) |

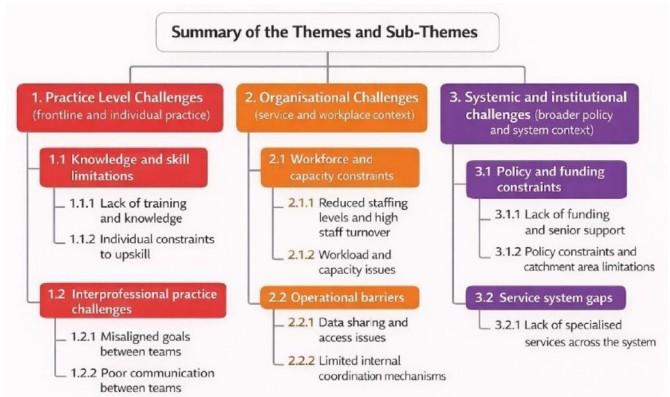

**Figure 1** Summary of the themes and subthemes representing barriers to integrated care.

development systems, where they reported limited opportunities to learn how to support families in TA, as well as how to collaborate with other professionals and understand their roles within the wider system:

> In terms of ensuring there is adequate care or adequate access for those who are homeless, no training whatsoever. (1132)

> We have a range of a range of expertise within the team, but nobody who's specifically qualified or trained to do work in that specific area…which can be stressful. *(3166)*

> No, no, not specifically [training], but they [homeless families] do, I would say, cover quite a large proportion of our caseload…One of the things that we find difficult [without training] is negotiating how it works within local authorities to get the support and the housing. And what is allowed and what isn't, and yeah it's a bit sticky, it's a bit of a minefield…these things change quite frequently as well, like they do for everything. *(3058)*

> From what I found when I've looked for training courses for my staff, there is not a lot widely available… especially not in the field of working with children who are facing multiple adversity. (2776)

Instead, teams often relied on informal, experience-based knowledge transfer:

> I think we've probably arrived at that [knowledge about supporting homeless children] by a mix of different things and a bit of learning on the job. But yeah, I'm not aware of anything specific that we've done… Young children especially tend to get a bit forgotten in the whole homelessness issue… there's a gap and obviously we've got our kind of health visitors and things like that, again, I wouldn't be confident about their level of understanding really when it comes to homelessness because it is a really complicated picture. (4048)

> I wouldn't say I have any specific training at all, it's more just experience and a general interest in

for children under 5 living in TA. Findings were organised using a multilevel framework that distinguishes between practice-level, organisational and systemic challenges (figure 1). This approach highlights how barriers operate at different levels of the service system and interact to shape participants' experiences.

### Practice level challenges (frontline and individual practice)
*Knowledge and skill limitations*
#### Lack of training and knowledge
The lack of accessible and relevant training was identified by participants as a significant barrier. Professionals highlighted a siloed approach within professional

looking after people who are I guess more socio economically deprived. (1388)

Moreover, this reported absence of training led to professionals stating they felt unprepared, possibly creating bottlenecks in service delivery particularly during the pandemic. As one participant reflected:

Our team, we did feel a bit out of our depth at one point. (7114)

### Individual constraints to upskill

Another significant reported barrier was the lack of time and capacity within roles to reflect or upskill which prevented professionals from moving beyond reactive, day-to-day duties to adopt a more strategic approach:

I think it [training] would be useful, it would be useful. I suppose in some sort of format, it's difficult to imagine how that would look for us as GPs, because I just don't know how that would look in terms of time, I suppose. (1388)

And then we're at the stage now we need a little bit more calm and capacity, a bit more head space. Twenty percent of our staff are off sick at the moment, so you try and put a training program together and it doesn't make sense. They're just on survival mode. So wait for the pandemic to die down and then the recovery period, probably in the spring summer, it's a little quieter, we will then put together, put our heads together. And look at ways in which we can address this. Either through some professional educational input, which goes to everybody, and we have to think again how to do it. But at the moment there's just no space, head space to do it. (4828)

### *Interprofessional practice challenges*
### Misaligned goals between teams

A fundamental challenge emerged around the differing priorities between teams, which reportedly hindered collaboration and reduced the effectiveness of integrated support. Professionals reported that while health professionals considered housing a critical determinant of health, housing officers focused solely on their legal duties:

Because for a housing officer, it's not really on their agenda, whether they've been for a smear test or whether the baby's had immunisations or whether or not that family is practicing safe sleep, all of these prevention things. It's not really their… it's not their priority. Their priority is to look at the legal duty that has been offered to that family… Housing Officers do not see that health is part of what will solve their problems. (1151)

This divergence in priorities was compounded by systemic fragmentation between teams:

We all got different funding. We're all different agencies and nobody talks to each other. And that's the

key problem. And it's like that with the NHS actually isn't it I mean that the left hand never talks to the right everything needs to come under one umbrella really when it comes to children because it's a whole process. You can't just have one thing being done. And the other bit nuts it's like having a jigsaw and not being able to finish it. (6555)

Furthermore, professionals described difficulties in securing buy-in from leadership across teams, reinforcing silos in service provision:

Sometimes you're asked during professional development or during your one to one's, what kind of professional development do you need, what will help you better to do your job properly. How can you help our residents better? And I would make suggestions and it's almost like well, we've tried that, we're not getting anywhere because the other team managers are not cooperating. So, it's almost like there's a barrier because we have tried…but you can't get the buy in from the other departments because there's a, it's just as bureaucracy and barriers… If people don't buy into a certain practice, and it comes from leadership really. (2226)

### Poor communication between teams

Limited communication was a recurring issue, reported as contributing to inefficiencies and delays in service provision. Many professionals described that they found themselves acting as intermediaries due to fragmented communication channels:

So then we become sort of mediators between what customer services cannot do and between back-office housing where they don't have that direct contact. Because even the phone lines and the emails, I think due to demand, in my opinion, it's almost like there's a barrier there. They can't access housing advisors, officers, housing officers directly. Unless sometimes by luck, the phone, somebody picks up or somebody decides they're going to email them. Because they might send them one email and six months later they hear back. But they are all irregular or poor communication. (2226)

the local authority decided to move these children from some of the homeless units to another homeless accommodation, either within the borough or outside the borough. So again, it took a long long time trying to trace these children because they disappear over, we don't know where they moved to. (1988)

Professionals also highlighted that escalation of their concerns via standard, recommended routes, such as commissioner intervention, was ineffective:

We've even had emails from commissioners saying oh if they do that, if they move somebody without letting you know, if you request for delayed dispersal and they disperse the person [then] this is the person to

email. And we continually do that but nothing happens, it still happens, it still happens yeah. (4029)

## Organisational challenges (service and workplace context)
### *Workforce and capacity constraints*
#### Reduced staffing levels and high staff turnover

Staffing shortages were highlighted as a significant barrier, with professionals describing burnout and high turnover rates as contributing to inefficiencies.

There's a lot of mandatory training you have to do to work for our trust. So the bank staff, we did hire, our recruitment time takes like, three, four months. It's a really, really long, arduous process. So that's the first part. And the second part where they've now asked, Does anyone want to do Saturdays or evenings? Everyone's burnt out—no. (4999)

Many teams were described as not able to access any additional funding or extra staff to help with the increased workload during the pandemic:

No. If anything it's less [funding], because you'd have to cover, as times gone on people are shielding or not shielding and all the rest of it. You're covering your colleagues work, staff are just ill, they're burned out, they're stressed. So staffing has been a huge, huge issue, much worse. And lots of people have been deployed as well. (5186)

#### Workload and capacity issues

High caseloads and limited capacity were key reported concerns, particularly in urgent care settings where referral processes created bottlenecks:

These urgent care centres will receive a walk in or a phone call from a patient asking to be seen, a child, and before even assessing the child they just send them straight to the community referral service. Which then creates a huge backlog for these community dental services. And it's just, it's very, very unreasonable when these children could actually be treated at the urgent care centres. (1132)

Participants described the backlog in services, which was exacerbated by the pandemic, as further restricting access to routine and preventative care:

Very difficult at the moment, because we have such a huge backlog because of the pandemic, not being able to see patients routinely for so long. We're still getting through our backlog, there are still patients who I've seen this week who haven't been seen for two years and they are registered in our books. So our registration is closed at the moment, so we're not taking on any new patients. So we can see them on an emergency basis, but not to take on to give them that preventative care, to give them that routine care. It's just to get them out of pain only. (1132)

### *Operational barriers*
#### Data sharing and access issues

Many professionals reported that data protection regulations, such as the UK General Data Protection Regulation, were often used as an excuse to restrict access to essential information about children in TA:

Internally, we just say GDPR is a b******* excuse because we're talking about safeguarding of vulnerable kids…So, if this was a domestic violence issue and a kid had been beaten up, everyone would know but we put these really vulnerable families in incredibly vulnerable situations in B and Bs with other kind of difficult single adults, or we put them out of borough, or we put them in disgusting accommodation, and suddenly, no one's really bothered… all we want to do is go, tell us who they are and we'll help them, and they just say, Oh no, it's GDPR… if we're talking about safeguarding children, then you need to share the data with us. (3597)

Participants felt that the consequence of this limitation in access to essential information meant staff often had to take on additional tasks outside their usual responsibilities to obtain the details needed to support families:

So I then have to do the secondary work of emailing other colleagues to say can you check the data, can you check if you've got any information for the landlord. Other than that, other teams might have access to the land registry, that tells us the type of property and who's the owner, whether they're a leaseholder or sub tenant etc. But again, you need a license for that, and only specific teams have that. (2226)

Moreover, the description indicated that existing data collection processes failed to capture key demographic details about children, limiting the ability to tailor support services effectively:

They don't even acknowledge the age of children, or the ethnicity of the family on their figures. One, because it's not important to them. Because if it was, it would be being measured… So these things we, we often will say we know how many people are in the household, but don't know the ages of the children. Where is that information being, because if you don't know how many children are in that situation, you can't change it. In that sense, that way the five year old needs to be in school, and that three year old need universal 16 hours of childcare provision, you know, to make sure there's some socialisation going on. (1151)

#### Limited internal coordination mechanisms

A further barrier identified was the lack of notification when families are relocated, which disrupts continuity of care. Participants were concerned that without proper alerts, existing data-sharing systems would fail to support

effective interagency collaboration, increasing the risk that vulnerable families are overlooked:

> The biggest barrier there is that no one tells any sort of health professional, any sort of social care professional when a family has declared themself homeless. Yeah, they will be placed in temporary accommodation and no one's doing any notification. (6631)

The impact of these data-sharing constraints was described as far-reaching, as many teams described still receiving referrals through chance conversations rather than structured communication channels:

> I did have a family that transferred into temporary accommodation and we did't know via housing or anything like that. It's only because the family nurse partnership team in that area contacted us and said oh did you know, and we were like okay, we'll take them on then. (3058)

Furthermore, participants reported that just as families are connected to support services, they are often relocated before they can fully benefit from them:

> They come in and they go out and by the time somebody finds them they're about to go out. And they might be in there for weeks on end of having no services. By the time, this is my colleagues, they've got the services up and running they're just about to move out so they're not there to receive all the lovely services. (1663)

Participants were concerned that ensuring continuity of care for these families largely depended on the dedication and goodwill of individual professionals rather than a reliable information sharing system.

> It just relies on the goodwill of the individual practitioner, because as I said during this pandemic I could have just not done anything. Or I could have done something that I thought was, in many ways, helpful to my clients. Yeah, so it's not something that I would have been kind of expected to, if I couldn't, I wouldn't have been expected to do, I didn't have to do it. (1988)

### Systemic and institutional challenges (broader policy and system context)
#### Policy and funding constraints
##### Lack of funding and senior support

Professionals highlighted how funding structures and reliance on senior support at the institutional level posed barriers to their ability to deliver integrated, preventative care for young children in TA. In particular, professionals highlighted the way service payment models in dentistry actively disincentivised early intervention:

> If you have a child under the age of two, you're not really going to be doing any treatment on them. So, the dentist, the dental practice doesn't get paid to have that preventative chat with the patients. So there's no financial incentive of seeing patients under the age of two, so I think that dentists don't see patients under the age of two. (1132)

In examples where effective integrated care was successful, participants noted that such initiatives were often dependent on charitable funding rather than sustained statutory investments:

> We got some money from the mayor's charity. So they gave us quite a considerable amount to help with trauma support work. They were doing, like, safeguarding training with the b&b staff and I kind of set up a little hub of connected professionals that all working together so we kind of were doing that kind of work, and then we've got money from the Morrison's CEO…to help homeless families in a large metropolitan area in the North of England. We provided hot meals in the B and Bs for a few months at the start of COVID. We did care packages we did, and cleaning packages and all that kind of stuff, all that money was paid on that. We commissioned a local restaurant to do hot food for all the families in b&bs. So I think we did 2000 meals over 3–4 months for all the families. And all that money kind of was, was spent on doing that. (3597)

> Our average house was decommissioned by the local authority, but we just felt as a charity it was a project that we just couldn't stop doing because it was always full. And there was a need there and we felt we're making a real difference for the families that did come through. And so we've been able to, you know, our boards very supportive of bridgehouse. So we were able to continue that and use some of our charity reserves and do some fundraising to make sure that that was able to continue. (4048)

Participants also highlighted that they felt that preventative care was hindered by a reliance on strategic leadership buy-in at the institutional level:

> Unless the officer, the managers and the, the higher, the chief officers, are aware that we need to do some prevention work to stop these families coming back into the homeless system—what we know is if babies and children are in the homeless system we know that they'll be more likely to be adults that will be within the homeless system. So it's really the open mindedness of those strategic leads to see that if we can implement and step in with prevention. (1151)

Another point raised was the potential success of homeless health visitors to integrating care; however, participants highlighted how policy specifications and a lack of senior-level support limited the adoption of such roles locally:

> So I think that, and I am actually trying to fight for it, for there to be a homeless self visitor for our area.

Particularly because of the Afghan refugees, we've suddenly got like 600 families all moved in. And they were very, our borough they were very against that, they weren't going to have it, but it's gone down really well in various areas including other boroughs. When I suggested it, they said it's not part of the spec for this area, so no. (5186)

### Policy constraints and catchment area limitations

Service provision was described as highly inconsistent due to catchment area policies:

So some children you just lose track of them because they don't ever get back…when it's impossible, for whatever reason, in exceptional circumstances we will retain that care just to keep the continuity going for the child. Despite the fact that it might be outside a contract of some sort. These are the constraints that really these poor families in a way are up against, institutional constraints. Not because the clinician doesn't want to see them but because the institution rules are such that they've now moved out, they need to go somewhere else… we've been prohibited by an institutional rule as opposed to what's clinically in the best interest of the child. (4828)

The data suggested that differences in eligibility criteria between local authorities and charities created frustration, as some individuals fell outside the support systems in place:

There's obviously cases where the local authority, particularly, that obviously they have an eligibility criteria, and some of the people that we support will at times fall outside of that where there can be frustration, I think, as a charity, because we kind of want to help anybody who's experiencing homelessness, and the local authority and some other charities in the city have different criteria around who they work with. (4048)

### Service system gaps
#### Lack of specialised services across the system

Participants were concerned that the absence of dedicated services for families in TA meant that existing services often failed to meet their needs:

I don't think one could be optimistic unless you redesign the service for those families. You have to understand their needs and I think health visitors are in a fantastic position to do that. And then you have to be able to define a response to those needs that actually is contracted specifically. It's not going to happen within the existing services, they will fail. We will fail these families, I know that. (4828)

Participants described specialist outreach and dedicated teams as highly effective in improving integrated care, yet their reflections also point to the absence of sufficient specialist capacity as an ongoing barrier to embedding these models more widely:

I'm very heavily involved in improving integrated care systems so that there's [specialist] outreach to general practice…And my experience of that has been brilliant because the parents absolutely love it. And I had a parent in tears, who had spent six years trying to get a specialist appointment…And it's professionally very satisfying because actually they all turn up, everybody knows where their GP is. And in terms of vulnerability and acceptability, its like they're on, you know, this is nearer to my home turf. There are lots of advantages but lots of disadvantages to the profession, in terms of travel time. (4828)

Health visitors are just leaving in droves and staffing issues are just getting worse and worse and worse. So this [having a specialist team] would be a good way to attract people into the profession and to retain the ones that are there. So I think having a specialist team would be really good. (5186)

## DISCUSSION

To our knowledge, this is one of only a few qualitative studies, set in a high-income country, which examines cross-sector professionals' experiences of delivering integrated care to children under 5 living in TA during the COVID-19 pandemic. This study builds on existing research highlighting barriers to accessing care faced by families from professionals' perspectives.[12] Many of the organisational and systemic barriers identified in this study, such as fragmented service provision, workforce instability, poor communication and limited data sharing, have been well documented in the literature.[20–23] These challenges, therefore, cannot be understood as direct consequences of the COVID-19 pandemic. Rather, the pandemic acted as a stress test that intensified pre-existing weaknesses, rendering long-standing structural failures more visible, and, in some cases, more acute. While the pandemic motivated innovative and opportunistic service integration to meet demands, it also exposed the fragility and inequity of existing systems, which had long lacked the infrastructure needed for sustainable collaboration. The findings of this study reveal a complex web of systemic challenges, including siloed team structures, inadequate communication, lack of specialised training, resource constraints and restrictive institutional policies, the majority of which remain barriers today.[41] This study highlights how these barriers operate across individual, team, work environment and institutional levels.

At the practice (individual) level, a key barrier was the lack of formal training for professionals, with many feeling unprepared and relying on experiential rather than structured evidence-based learning. Similar to previous research,[42 43] the absence of training resources increased staff stress and lowered confidence. Limited professional development confined staff to reactive crisis

management, reducing capacity for sustainable support and skill-building. This reactive focus aligns with previous research, which found that services working with vulnerable populations are frequently crisis-driven and tend to diminish once immediate risks subside.[44] Embedding tailored training into routine practice is essential to improve service continuity and workforce preparedness.[45] While inadequate training and reliance on experiential learning are long-standing issues in service provision, the pandemic intensified their consequences by accelerating redeployment, increasing caseload complexity and reducing access to informal learning and supervision, thereby amplifying professional stress and reactive practice.

At the organisational and team level, conflicting priorities and poor communication, particularly between health and housing services, were significant barriers to integrated care, often sidelining the child's needs. These challenges led to inefficiencies, delays and fragmented services, consistent with previous findings on cross-sector collaboration.[44 46] Professionals often took on extra coordination tasks to bridge communication gaps and prevent families from falling through service gaps, increasing their workload and detracting from direct service delivery - a response similarly noted in a study on healthcare for vulnerable populations.[47] Staffing shortages further disrupted service continuity, hindered relationship-building and weakened collaboration, echoing existing research highlighting the impacts of workforce instability.[48] Although workforce instability and high turnover were well-recognised challenges prior to the pandemic,[20–23] COVID-19 exacerbated these trends through staff illness, burnout, emergency redeployment and rapid organisational changes, significantly reducing opportunities for relationship-building and coordinated working. The pandemic further strained already understaffed services,[49] leaving professionals in a constant state of 'firefighting' and contributing to widespread burnout.[50 51] Postpandemic challenges persist, with housing providers frequently hiring at pace to fill urgent gaps, often prioritising speed over skills and experience.[52] This reactive approach not only affects service quality but also makes it harder to foster the stable, long-term professional relationships needed for effective, child-centred care.

Within the work environment, workload pressures led to teams operating in silos, with poor collaboration creating inefficiencies and backlogs. This often resulted in delayed referrals and missed opportunities for children in TA to access vital health and support services. While siloed working across housing and social sectors supporting homeless populations is well-documented,[18 53 54] this study demonstrates how pandemic-related pressures worsened not only cross-sector fragmentation, but also fragmentation within healthcare teams themselves, highlighting a less frequently examined barrier to integrated care for children in TA.

One of the most significant operational barriers to integrated care was the restriction of data sharing between teams, limited internal coordination mechanisms and thus the inability to identify and track access and outcomes over time and geography. The lack of interoperable IT systems is a well-documented challenge for integrated homeless services, limiting cross-sector collaboration.[55] A 2015 survey by the Health Service Journal found that over 60% of professionals believed that data protection regulations were hindering their efforts to integrate health and social care services.[56] Our findings also reveal that homeless children are often excluded from data-sharing structures designed to safeguard vulnerable groups, highlighting how current systems overlook their needs—an issue made especially visible during the pandemic, when there was a greater reliance on digital systems, and during a time of heightened risk for children.[57] Greater legislative flexibility is needed to enable secure, efficient information sharing,[44] especially for children in their formative years, as their frequent relocations increase the risk of them slipping through the gaps in care.[58 59] Enabling local systems to link and share data, professionals can gain a more complete picture of the health and social needs of homeless children.[60]

At the systemic level, funding constraints, particularly limited investment in prevention, were noted as a barrier to integrated care, aligning with existing research.[61–63] Children under 5 in TA, already at a disadvantage, feel the impact of this most acutely, with the pandemic further widening health inequalities.[12] The lack of investment in prevention not only undermines children's long-term health outcomes but also places greater strain on already overstretched services.[63] System-level differences in catchment areas and eligibility criteria among services posed further challenges to integrated care. These inconsistencies often lead to unnecessary duplication of services, as highlighted by the Institute for Government which found that overlapping responsibilities can result in inefficiencies.[30] More critically, this study found that such misalignments frequently disrupt continuity of care, causing vulnerable families to fall through the gaps in siloed support systems. These findings suggest that without targeted post-pandemic reform to address long-standing systemic barriers, rather than temporary crisis responses, services risk reverting to prepandemic patterns of fragmented care, with enduring consequences for children under 5 living in TA.

## Strengths

A key strength of this study was the combination of diverse cross-sector perspectives, capturing insights from professionals across health, housing and social care. The national scope offers broad applicability across England and strengthens the transferability of the findings beyond local contexts. The use of semistructured interviews and detailed topic guides that were informed by the larger team's expertise in the area provided consistency across participants while allowing in-depth exploration of individual experiences, gathering rich nuanced insights into barriers faced by professionals. The research fellow holds

a PhD in sociology and is well trained in the use of qualitative research methods. She was closely supervised by lead researchers on the project with in-depth contextual and methodological expertise relevant to this research. Additionally, the use of co-coding and consensus discussions during analysis enhanced the reliability and rigour of the findings, reducing individual bias and ensuring consistency in theme identification.[64] The research also aligns with current national and local priorities, contributing to the recent all-party parliamentary group call for evidence on TA and supporting National Health Service integrated care boards' efforts to address health inequalities and homelessness.[65] Finally, the study was supported by a team with significant expertise in health and social care systems research, strengthening the rigour and interpretation of the findings.

### Limitations

A limitation of this study is that recruitment was particularly challenging as the study took place during the COVID-19 pandemic, when there were significant service disruptions, staff redeployment and heightened professional responsibilities. Therefore, the availability of some staff was limited, which extended the recruitment timeframe and localities of where professionals were recruited from—relying on opportunistic sampling rather than systemic representation of service provision across England. As a result, the study does not account for local variation in service provision or allow for comparative analysis between regions. However, data saturation was reached, indicating that key themes were consistently identified across interviews despite recruitment challenges. Lastly, the data was collected almost 3 years ago; however, the findings remain valuable for informing current and future practices, particularly as services continue to recover from the pandemic and many of the barriers remain relevant.

### Recommendations

Strengthening integrated care for children in TA requires targeted cross-sector training to build understanding of population needs and different service roles, improving collaboration and delivery. In addition, a notification system, as proposed by the Shared Health Foundation,[57] could alert services when families enter TA, helping prevent children from falling through gaps. In the short term, investment in key workers, shown effective in the Supporting Families programme,[66] would support continuity of care for families facing relocations. Ultimately, insights from this study highlight the urgent need for substantial capital investment in integrated recovery programmes and the adoption of a universal theory-driven model of integrated care that prioritises cross-sector collaboration, early intervention and preparedness for future crises.

### Future research

Future research should focus on a longer term post-COVID study to reflect on pandemic learnings and further examine persistent barriers to cross-sector collaboration in supporting children in TA. Future research should also explore how geographical differences affect the delivery and accessibility of support for children in TA. Local variation in resources, services and inter-agency coordination may significantly influence outcomes and should be examined through comparative, place-based analysis. Lastly, further research focusing on the experiences and outcomes of families themselves would provide valuable insights into the real-world impacts of systemic barriers and help shape more child-centred policies.

## CONCLUSIONS

This study highlights the complex, multilevel barriers that professionals face when delivering integrated care to children under 5 in TA. From siloed working practices and limited training to staffing shortages and restrictive data-sharing policies, these challenges collectively hinder cross-sector collaboration and reflect long-standing issues in service delivery. The findings highlight the urgent need for systemic reforms, including improved cross-sector communication and training initiatives, flexible data-sharing frameworks and greater investment in integrated recovery plans to support children postpandemic. Leveraging cross-sector expertise and fostering interdisciplinary collaboration can cultivate a more skilled, coordinated workforce, ensuring equitable access to quality care for children under 5 in TA.

**Author affiliations**
[1]Medical School, University College London Faculty of Medical Sciences, London, UK
[2]Applied Social Sciences, De Montfort University Faculty of Health and Life Sciences, Leicester, UK
[3]Population, Policy and Practice Research and Teaching Department, UCL GOS ICH, London, UK
[4]Flinders University College of Nursing and Health Sciences, Bedford Park, South Australia, Australia
[5]School of Nursing & Midwifery, Flinders University, Adelaide, South Australia, Australia
[6]Specialist Children's and Young People's Services, East London NHS Foundation Trust, London, UK
[7]Bartlett School of Graduate Studies, University College London, London, UK
[8]Community Paediatrics, Nottingham University Hospitals NHS Trust, Nottingham, UK

**Acknowledgements** The authors wish to thank all of the professionals who took the time out of their busy schedules to take part in these interviews, especially given the challenging circumstances in which data were collected. We would like to acknowledge the hard work of researchers that supported the development of the project ethics and data collection—Dr Margarita Garfias Royo and Dr Kriss Fearon, respectively. We would also like to thank all our partner organisations who supported CHAMPIONS and all the participants who kindly gave their time to support the study. We would also like to thank Dr Diana Margot Rosenthal for her contribution to the conceptualisation of the project, writing of the CHAMPIONS project proposal, and sharing of insights from her PhD research on health and healthcare service access for children under 5 living in temporary accommodation due to experiencing homelessness. Finally, the authors would like to thank UKRI/ESRC for funding this important work.

**Contributors** PASP: data curation, analysis, writing (original draft preparation). NS: conceptualisation, funding acquisition, methodology, supervision, analysis, writing (review and editing). HA: analysis, writing (original draft preparation). SA: supervision, data interpretation, writing (review and editing). MA: data interpretation, writing (review and editing). YKP: data interpretation, writing (review

and editing). MH: conceptualisation, funding acquisition, writing (review and editing). MU: conceptualisation, funding acquisition, writing (review and editing). ML: conceptualisation, funding acquisition, project administration, methodology, supervision, analysis, writing (review and editing), guarantor. ML is the senior author.

**Funding** ESRC, as part of UK Research & Innovation's rapid response to COVID-19 (No.ES/V016253/1).

**Competing interests** None declared.

**Patient and public involvement** Patients and/or the public were not involved in the design, or conduct, or reporting, or dissemination plans of this research.

**Patient consent for publication** Not applicable.

**Ethics approval** This study involves human participants and ethics approval was obtained by the UCL Research Ethics Committee (ID number: 9277.004). Participants gave informed consent to participate in the study before taking part.

**Provenance and peer review** Not commissioned; externally peer reviewed.

**Data availability statement** Data are available on reasonable request. The data that support the findings of this study are available from the corresponding author on reasonable request.

**ORCID iDs**
Poppy Angelica Spaceman Pierce https://orcid.org/0009-0007-0071-1469
Nadzeya Svirydzenka https://orcid.org/0000-0003-3085-6785
Shereen Allaham https://orcid.org/0000-0003-0275-3228
Matthew Ankers https://orcid.org/0000-0002-4246-8010
Michelle Heys https://orcid.org/0000-0002-1458-505X
Marcella Ucci https://orcid.org/0000-0001-5618-7247
Monica Lakhanpaul https://orcid.org/0000-0002-9855-2043

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
