## [Reviewer comments · BMJ Open]

ARTICLE DETAILS

Title (Provisional)

Exploring barriers to integrated care for children under five living in temporary accommodation: a qualitative study of professionals' experiences during the COVID-19 pandemic in England

Authors

Pierce, Poppy Angelica Spaceman; Svirydzenka, Nadzeya; Adil, Haleema; Allaham, Shereen; Ankers, Matthew; Parry, Yvonne Karen; Heys, Michelle; Ucci, Marcella; Lakhanpaul, Monica

VERSION 1 - REVIEW

Reviewer	1
Name	Cochliou, Despina
Affiliation	University of Nicosia
Date	19-Aug-2025
COI	None

Thank you for this paper. It was really interesting reading it. I have only some suggestions, which I think will improve the paper even more. The research findings have identified a series of organisational difficulties regarding health, housing and social services, which are not immediate consequences of the pandemic. These difficulties had already been discussed and researched 20 years ago for both adult and children services alike. Therefore, it would have been important to review the existing literature on these systemic failures and identify separately those that arise as a result of the pandemic. Setting a robust background for the study with existing literature would enable a more refined interpretation of the findings.

Reviewer	2
Name	Katz, Ilan
Affiliation	University of New South Wales, Social Policy Research Centre, Faculty of Arts and Social Science
Date	07-Dec-2025

COI

None

The paper describes the barriers to effective service delivery for children in temporary accommodation from the perspective of service providers in England. Overall this is a well written article which adequately describes a number of challenges for practitioners to integrate different services, as is necessary for this cohort of service users.

The article would be strengthened if the findings were presented according to a clear analytic framework - eg separating practice issues, organisational challenges and systemic challenges. The discussion could then also be organised in this way.

A second issue is that it is not always clear what the specific contribution of COVID 19 has been to service provision for these children, given that the barriers described are very similar to barriers to integrated/joined up services in a range of different areas which have been described in the literature over several decades. What is the specific contribution of this research to the evidence base in this area, in particular how has the pandemic affected these trends and what are the implications post-pandemic?

It would also strengthen the article if a couple of examples of good practice were also included in the findings, given, as indicated above, that the barriers to integrated service provision are very well known.

VERSION 1 - AUTHOR RESPONSE

Reviewer 1	
The research findings have identified a series of organisational difficulties regarding health, housing and social services, which are not immediate consequences of the pandemic. These difficulties had already been discussed and researched 20 years ago for both adult and children services alike. Therefore, it would have been important to review the existing literature on these systemic failures and identify separately those that arise as a result of the pandemic. Setting a robust background for the study with existing literature would enable a more refined interpretation of the findings.	Thank you for raising this. This comment has now been addressed by expanding the introduction literature review and adding further detail in the discussion, in order to incorporate existing literature on long-standing systemic challenges and to clearly distinguish these from difficulties arising specifically as a result of the pandemic. Please see amendments below: Sentences added to introduction: (2nd paragraph): Long-standing research has consistently identified structural barriers to

effective health, housing, and social care provision for homeless families, well before the COVID-19 pandemic. Studies spanning the past two decades highlight persistent challenges including fragmented service delivery, limited cross-sector coordination, workforce instability, and bureaucracies across systems supporting both adults and children experiencing homelessness (20-23).

Last paragraph: While previous research has extensively documented systemic barriers to integrated care for homeless populations, less is known about how these long-standing challenges were experienced, adapted to, or exacerbated during the COVID-19 pandemic.

Sentences added to discussion:

1st paragraph: Many of the organisational and systemic barriers identified in this study, such as fragmented service provision, workforce instability, poor communication, and limited data sharing, have been well documented in the literature (20-23). These challenges therefore cannot be understood as direct consequences of the COVID-19 pandemic. Rather, the pandemic acted as a stress test that intensified pre-existing weaknesses, rendering long-standing structural failures more visible, and, in some cases, more acute.

2nd paragraph: While inadequate training and reliance on experiential learning are long-standing issues in service provision, the pandemic intensified their consequences by accelerating redeployment, increasing caseload complexity, and reducing access to informal learning and supervision, thereby amplifying professional stress and reactive practice.

3rd paragraph: Although workforce instability and high turnover were well-recognised challenges prior to the pandemic (20-23), COVID-19 exacerbated these trends

	through staff illness, burnout, emergency redeployment, and rapid organisational changes, significantly reducing opportunities for relationship-building and coordinated working. 4th paragraph: While siloed working across housing and social sectors supporting homeless populations is well-documented (18,49,50), this study demonstrates how pandemic-related pressures worsened not only cross-sector fragmentation, but also fragmentation within healthcare teams themselves, highlighting a less frequently examined barrier to integrated care for children in TA. Fifth paragraph: Our findings also reveal that homeless children are often excluded from data-sharing structures designed to safeguard vulnerable groups, highlighting how current systems overlook their needs - an issue made especially visible during the pandemic, when there was a greater reliance on digital systems, and during a time of heightened risk for children (57). Sixth paragraph: These findings suggest that without targeted post-pandemic reform to address long-standing systemic barriers, rather than temporary crisis responses, services risk reverting to pre-pandemic patterns of fragmented care, with enduring consequences for children under 5 living in TA.
Reviewer 2	
The article would be strengthened if the findings were presented according to a clear analytic framework - eg separating practice issues, organisational challenges and systemic challenges. The discussion could then also be organised in this way.	Thank you for your helpful suggestion. The findings and discussion have now been reorganised using a clear analytic framework that distinguishes between practice issues, organisational challenges, and systemic

	challenges. Please see the sentences and sub-headings used in the results section below: Findings were organised using a multi-level framework that distinguishes between practice-level, organisational, and systemic challenges (Figure 1). This approach highlights how barriers operate at different levels of the service system and interact to shape participants' experiences. 1. Practice level challenges (frontline and individual practice) 1.1 Knowledge and skill limitations 1.1.1 Lack of training and knowledge 1.1.2 Individual constraints to upskill 1.2 Interprofessional practice challenges 1.2.1 Misaligned goals between teams 1.2.2 Poor communication between teams 2. Organisational challenges (service and workplace context) 2.1. Workforce and capacity constraints 2.1.1 Reduced staffing levels and high staff turnover 2.1.2 Workload and capacity issues 2.2 Operational barriers 2.2.1 Data sharing and access issues 2.2.2 Limited internal coordination mechanisms 3. Systemic and institutional challenges (broader policy and system context) 3.1. Policy and funding constraints 3.1.1 Lack of funding and senior support 3.1.2 Policy constraints and catchment area limitations 3.2 Service system gaps 3.2.1 Lack of specialised services across the system
A second issue is that it is not always clear what the specific contribution of COVID 19 has been to service provision for these children, given that the barriers described are very similar to barriers to integrated/joined up services in a range of different areas which have been described in the literature over	Thank you for this important point. We have now clarified the specific contribution of COVID-19 by explicitly distinguishing pre-existing well-documented barriers to integrated service provision from those that were exacerbated, or newly emerged from during the pandemic. The discussion has

several decades. What is the specific contribution of this research to the evidence base in this area, in particular how has the pandemic affected these trends and what are the implications post-pandemic?

been revised to highlight how COVID-19 intensified existing trends (e.g. service fragmentation, capacity constraints, and access barriers), and reshaped cross-sector working. We also now more clearly articulate the post-pandemic implications of these findings and the added contribution of this study to the existing evidence base.

Sentences added to introduction:

(2nd paragraph): Long-standing research has consistently identified structural barriers to effective health, housing, and social care provision for homeless families, well before the COVID-19 pandemic. Studies spanning the past two decades highlight persistent challenges including fragmented service delivery, limited cross-sector coordination, workforce instability, and bureaucracies across systems supporting both adults and children experiencing homelessness (20-23).

Last paragraph: While previous research has extensively documented systemic barriers to integrated care for homeless populations, less is known about how these long-standing challenges were experienced, adapted to, or exacerbated during the COVID-19 pandemic.

Sentences added to discussion:

1st paragraph: Many of the organisational and systemic barriers identified in this study, such as fragmented service provision, workforce instability, poor communication, and limited data sharing, have been well documented in the literature (20-23). These challenges therefore cannot be understood as direct consequences of the COVID-19 pandemic. Rather, the pandemic acted as a stress test that intensified pre-existing weaknesses, rendering long-standing structural failures more visible, and, in some cases, more acute.

2nd paragraph: While inadequate training and reliance on experiential learning are

long-standing issues in service provision, the pandemic intensified their consequences by accelerating redeployment, increasing caseload complexity, and reducing access to informal learning and supervision, thereby amplifying professional stress and reactive practice.

3rd paragraph: Although workforce instability and high turnover were well-recognised challenges prior to the pandemic (20-23), COVID-19 exacerbated these trends through staff illness, burnout, emergency redeployment, and rapid organisational changes, significantly reducing opportunities for relationship-building and coordinated working.

4th paragraph: While siloed working across housing and social sectors supporting homeless populations is well-documented (18,49,50) this study demonstrates how pandemic-related pressures worsened not only cross-sector fragmentation, but also fragmentation within healthcare teams themselves, highlighting a less frequently examined barrier to integrated care for children in TA.

Fifth paragraph: Our findings also reveal that homeless children are often excluded from data-sharing structures designed to safeguard vulnerable groups, highlighting how current systems overlook their needs - an issue made especially visible during the pandemic, when there was a greater reliance on digital systems, and during a time of heightened risk for children (57).

Sixth paragraph: These findings suggest that without targeted post-pandemic reform to address long-standing systemic barriers, rather than temporary crisis responses,

	services risk reverting to pre-pandemic patterns of fragmented care, with enduring consequences for children under 5 living in TA.
It would also strengthen the article if a couple of examples of good practice were also included in the findings, given, as indicated above, that the barriers to integrated service provision are very well known.	We have now included some examples of effective integrated care within the findings, focusing on what facilitated good practice as well as the barriers to sustaining it, reflecting the experiences and perspectives shared by participants. Added sentences and quotes to Results section: 3.1. Policy and funding constraints 3.1.1 Lack of funding and senior support In examples where effective integrated care was successful, participants noted that such initiatives were often dependent on charitable funding rather than sustained statutory investments: “We got some money from the mayor’s charity. So they gave us quite a considerable amount to help with trauma support work. They were doing, like, safeguarding training with the b&b staff and I kind of set up a little hub of connected professionals that all working together so we kind of were doing that kind of work, and then we’ve got money from the Morrison’s CEO...to help homeless families in a large metropolitan area in the North of England. We provided hot meals in the B and Bs for a few months at the start of COVID. We did care packages we did, and cleaning packages and all that kind of stuff, all that money was paid on that. We commissioned a local restaurant to do hot food for all the families in b&bs. So I think we did 2000 meals over 3-4 months for all the families. And all that money kind of was, was spent on doing that.” (3597)

“Our average house was decommissioned by the local authority, but we just felt as a charity it was a project that we just couldn't stop doing because it was always full. And there was a need there and we felt we're making a real difference for the families that did come through. And so we've been able to, you know, our boards very supportive of bridgehouse. So we were able to continue that and use some of our charity reserves and do some fundraising to make sure that that was able to continue.” (4048)

Another point raised was the potential success of homeless health visitors to integrating care; however, participants highlighted how policy specifications and a lack of senior-level support limited the adoption of such roles locally:

“So I think that, and I am actually trying to fight for it, for there to be a homeless self visitor for our area. Particularly because of the Afghan refugees, we've suddenly got like 600 families all moved in. And they were very, our borough they were very against that, they weren't going to have it, but it's gone down really well in various areas including other boroughs. When I suggested it, they said it's not part of the spec for this area, so no.” (5186)

3.2. Service system gaps

3.2.1 Lack of specialised services across the system

Participants described specialist outreach and dedicated teams as highly effective in improving integrated care, yet their reflections also point to the absence of

	sufficient specialist capacity as an ongoing barrier to embedding these models more widely: “I’m very heavily involved in improving integrated care systems so that there’s [specialist] outreach to general practice...And my experience of that has been brilliant because the parents absolutely love it. And I had a parent in tears, who had spent six years trying to get a specialist appointment...And it’s professionally very satisfying because actually they all turn up, everybody knows where their GP is. And in terms of vulnerability and acceptability, it’s like they’re on, you know, this is nearer to my home turf. There are lots of advantages but lots of disadvantages to the profession, in terms of travel time.” (4828) “Health visitors are just leaving in droves and staffing issues are just getting worse and worse and worse. So this [having a specialist team] would be a good way to attract people into the profession and to retain the ones that are there. So I think having a specialist team would be really good.” (5186)
--	---

VERSION 2 - REVIEW

Reviewer 1

Name Cochliou, Despina

Affiliation University of Nicosia

Date 12-Jan-2026

COI

Thank you for resubmitting! Congratulations for this publication.